# Pregnancy outcomes after hysteroscopic surgery in women with cesarean scar syndrome

Shunichiro Tsuji[1]◐*, Akimasa Takahashi[1]◐, Asuka Higuchi[1], Akiyoshi Yamanaka[1], Tsukuru Amano[1], Fuminori Kimura[1], Ayumi Seko-Nitta[2], Takashi Murakami[1]

1 Department of Obstetrics and Gynecology, Shiga University of Medical Science, Seta Tsukinowa-cho, Otsu City, Shiga, Japan, 2 Department of Radiology, Shiga University of Medical Science, Seta Tsukinowa-cho, Otsu City, Shiga, Japan

◐ These authors contributed equally to this work.
* tsuji002@belle.shiga-med.ac.jp

**Data Availability Statement:** All relevant data are within the paper and its Supporting Information files.

## Abstract

Cesarean scar defect often causes postmenstrual abnormal uterine bleeding, dysmenorrhea, chronic pelvic pain, and infertility, which are collectively known as cesarean scar syndrome (CSS). Several studies have reported that hysteroscopic surgery can restore fertility in women with CSS. The study aimed to identify factors that influence subsequent pregnancy following hysteroscopic surgery. Therefore, we studied 38 women with secondary infertility due to CSS who underwent hysteroscopic surgery at Shiga University of Medical Hospital between July 2014 and July 2019. Our hysteroscopic procedure included inferior edge resection and superficial cauterization of the cesarean scar defect under laparoscopic guidance. Patients were followed up for 3 to 40 months after surgery. Surgery was successful in all cases and no complications were observed. Twenty-seven patients (71%) became pregnant (pregnant group), while 11 (29%) did not (non-pregnant group). Baseline characteristics of age, body mass index, gravidity, parity, previous cesarean section, presence of endometriosis, retroflex uterus, and preoperative residual myometrial thickness were not significantly different between the groups. However, the median residual myometrium thickness was significantly higher after surgery than before surgery in the pregnant group (1.9 [1.1–3.6] vs 4.9 [3.4–6.6] mm, P<0.0001), whereas this difference was not significant in the non-pregnant group. Of those who became pregnant, 85% conceived within 2 years of surgery. Although three pregnancies resulted in abortion and one is ongoing at the time of writing, 23 pregnancies resulted in healthy babies at 35–38 gestational weeks by scheduled cesarean sections with no obstetrical complications due to hysteroscopic surgery. The average birth weight was 3,076 g. Our findings support that hysteroscopic surgery is a safe and effective treatment for secondary infertility due to CSS. The thickness of the residual myometrium may be a key factor that influences subsequent pregnancy in women with CSS.

**Funding:** This study was supported by the Grants-in-Aid for Scientific Research (KAKENHI; 20K09616). The grant provided financial support for preparation of the article, such as English language editing services and Open Access Publication Fee. https://www.jsps.go.jp/english/index.html.

**Competing interests:** The authors have declared that no competing interests exist.

## Introduction

The rate of cesarean section is increasing worldwide. In Japan, it reached 18.5% in 2013, which is nearly a two-fold increase in the past two decades [1]. With the increase in cesarean sections, the incidence of cesarean section scar-related complications has also risen. Cesarean section often results in a cesarean scar defect (CSD), also known as isthmocele, which reportedly occurs in 24–84% of women after cesarean section [2,3]. Although there is still no clear universal definition of this term, it is commonly used in the literature to indicate a myometrial discontinuity or a hypoechoic region in the lower anterior uterine wall via transvaginal ultrasound detection [3,4]. CSD can cause secondary infertility with postmenstrual abnormal uterine bleeding, dysmenorrhea, and chronic pelvic pain [5]. These symptoms are collectively known as cesarean scar syndrome (CSS) [6]. We have previously investigated the management of secondary infertility for patients with CSS in Japan and found that surgical treatment, including laparotomy, laparoscopy, and hysteroscopy, was effective for restoring fertility in such patients [5]. Although there is an ongoing debate regarding the best surgical approach, hysteroscopic treatment is considered less invasive than other approaches and may be an effective treatment option for restoring fertility in women with CSS [7–9]. However, the predictors of subsequent reproductive outcomes following hysteroscopic treatment are still not clear.

Therefore, we aimed to investigate the reproductive and obstetric outcomes and the interval between treatment and conception following hysteroscopic procedures.

## Materials and methods

### Study population and recruitment

The participants of this retrospective study were patients who underwent hysteroscopic surgery for CSS between July 2014 and July 2019 at the Shiga University of Medical Science. The inclusion criteria were women diagnosed with CSS. CSS was diagnosed by detecting both CSD and the presence of abnormal uterine bleeding or liquid pooling on transvaginal ultrasonography. Exclusion criteria were women who were not intending to conceive. All patients underwent cancer screenings to exclude abnormal cervical cytology. Subsequent pregnancy was confirmed by the presence of a gestational sac in the uterus. Participants were categorized into two groups, pregnant or non-pregnant, according to the outcome of conception after surgery. Written informed consent was obtained from all patients prior to surgery. All data were fully anonymized before we accessed them, and patients' medical records were accessed between June 2020 and July 2020. This study was approved by the Ethics Committee at Shiga University of Medical Science (approval number; R2020-039) and performed at the Shiga University of Medical Science.

### Surgical procedures

We performed hysteroscopic surgery as previously described [10]. Briefly, hysteroscopic surgery was performed using a rigid 30˚ hysteroscope (4 mm telescope) and working elements (#27050, KARL STORZ, Germany) connected to a video camera and monitor (Olympus, Tokyo, Japan). Diagnostic laparoscopy was performed simultaneously to monitor accidental perforation at the site of the CSD and to treat other causes of infertility, such as endometriosis, because the aim of this operation was to restore fertility. Cervical dilation was carried out the day before surgery. First, hysteroscopic resection of the CSD inferior edge was performed using a cutting loop electrode to enable visualization of the diverticulum. Next, the entire CSD was cauterized using a ball electrode (Fig 1 and S1 File). Patients were discharged 2 to 3 days after surgery.

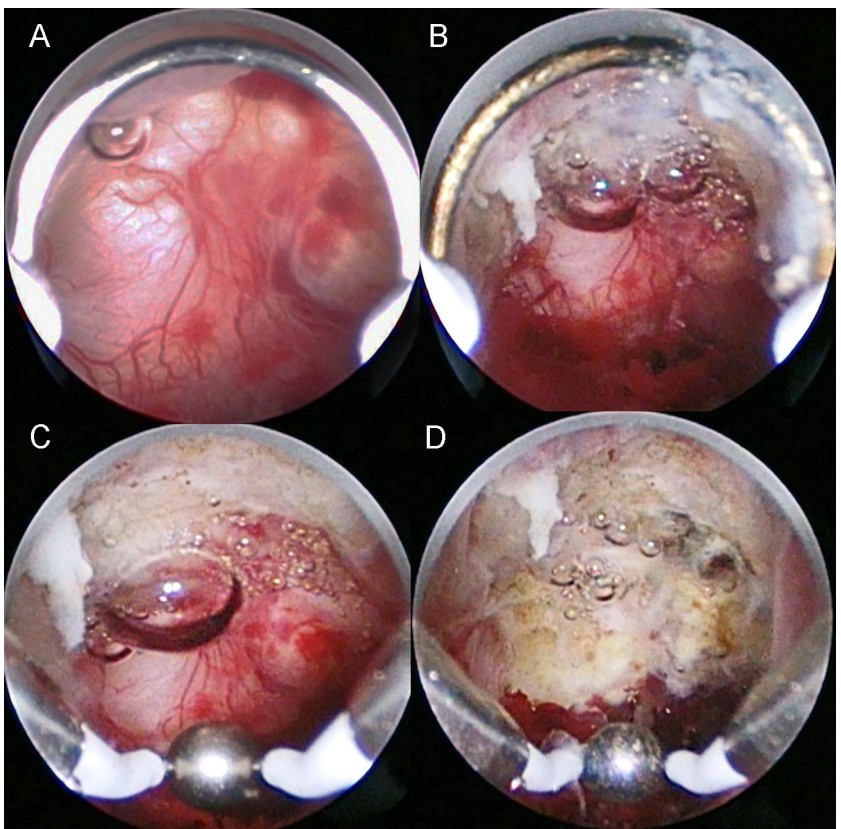

**Fig 1. Intraoperative images of the hysteroscopic surgery procedure.** (A) Abnormal hypervascularity is observed in the cesarean scar defect. (B) Cutting of the inferior edge of the cesarean scar defect using a cutting loop electrode. (C) Cauterization of all areas including the abnormal vasculature in the cesarean scar defect. (D) Appearance after cauterization using a ball electrode.

## Data collection

Baseline characteristics included age, body mass index, gravidity, parity, previous cesarean section(s), frequency of endometriosis, and retroflex uterus. Residual myometrial thickness (RMT) was measured by magnetic resonance imaging (MRI) before and 2 months after surgery using a 1.5-T instrument (SIGNA HDxt; GE Healthcare Waukesha, WI, USA) with a cardiac coil. The settings for MRI were applied as described previously [10]. All measurements were conducted by one senior radiologist using a high-resolution monitor. The interval from operation to conception was evaluated based on patient medical records. For patients who did not continue to attend our hospital, we confirmed their current situation via a medical information provision form from their referral hospital or by telephone.

## Statistical analysis

All data were analyzed using GraphPad Prism ver.7 (GraphPad Software, Inc., San Diego, CA, USA). The D'Agostino-Pearson test was used to evaluate data distribution. Normally distributed data are expressed as mean ± standard deviation. Data with a non-normal distribution are presented as median (interquartile range). Categorical data were compared by Fisher's exact test. Comparisons between the pregnant and non-pregnant participants were carried out using an unpaired two-tailed $t$-test or the Mann-Whitney U test for parametric and non-

parametric data, respectively. The cumulative pregnancy rate was evaluated using the Kaplan-Meier method. Statistical significance was defined as P < 0.05 in all cases.

## Results

Thirty-eight patients who met the inclusion criteria during the study period were included in the analysis. No complications were experienced by any patients. Twenty-seven patients (71%) became pregnant (pregnant group) and 11 patients (29%) did not become pregnant (non-pregnant group) (S1 Table). Baseline characteristics did not significantly differ between the two groups (Table 1). Two cases did not undergo laparoscopy for individual reasons; therefore, information on endometriosis was not available at the time of hysteroscopic surgery. Endometriosis was detected in 19 patients (52.8%) during hysteroscopic surgery. A blueberry spot was detected in one case on the surface of the CSD (S1 Fig).

Pre- and postoperative RMTs were measured in 34 patients (postoperative measurement was not carried out in four cases due to individual reasons). The median pre- and postoperative RMT measurements were 2.0 (1.1–3.7) mm and 4.4 (2.5–6.0) mm, respectively (P < 0.0001) (Fig 2A). Considering the pregnant group alone, the postoperative RMT was significantly higher than the preoperative RMT (1.9 [1.1–3.6] mm vs 4.9 [3.4–6.6] mm, P < 0.0001); however, the difference was not significant in the non-pregnant group (2.1 [0.8–3.9] mm vs 2.3 [2.1–4.4] mm) (Fig 2B and 2C). A significant difference was observed in postoperative RMT between the pregnant and non-pregnant groups (4.9 [3.4–6.6] mm vs 2.3 [2.1–4.4] mm, respectively; P = 0.02).

After hysteroscopy, 17 patients became pregnant within 1 year and six became pregnant during the following year. The cumulative pregnancy rate is illustrated in Fig 3. Pregnancy is ongoing in one case and three cases resulted in spontaneous abortions. The mean birth weight among all the patients who gave birth was 3,076 ± 435 g. One patient underwent a scheduled cesarean section at 35 gestational weeks due to placenta previa. All other deliveries were scheduled cesarean sections following the individual policies of the obstetric hospitals; four delivered at 36 gestational weeks, eight delivered at 37 gestational weeks, and 10 delivered at 38 gestational weeks. No severe obstetrical complications, such as uterine rupture, occurred up to the day of cesarean section in any case.

## Discussion

To the best of our knowledge, this is the first report to evaluate the changes in residual myometrium thickness after surgery in relation to subsequent reproductive outcomes. The present

**Table 1. Comparison of patients and clinical data.**

| | Pregnancy (n = 27) | Non-pregnancy (n = 11) | P |
|---|---|---|---|
| Age, yrs | 35.6±3.4 | 37.0±4.2 | n.s. |
| BMI | 22.2±3.7 | 21.5±3.1 | n.s. |
| Gravidity | 1 (1–2) | 2 (1–2) | n.s. |
| Parity | 1 (1–1) | 1 (1–2) | n.s. |
| Previous CS | 1 (1–1) | 1 (1–2) | n.s. |
| Endometriosis (%) | 14 (52)* | 5 (45) | n.s. |
| Retroflexion(%) | 14(52) | 3(27) | n.s. |
| RMT preoperatively (mm) | 2.3(1.3–3.8) | 2.1 (0.8–3.9) | n.s. |

BMI: Body mass index, CS: cesarean section, RMT: Residual myometrium thickness, Data are median (quartiles)

*Two patients did not undergo laparoscopy.

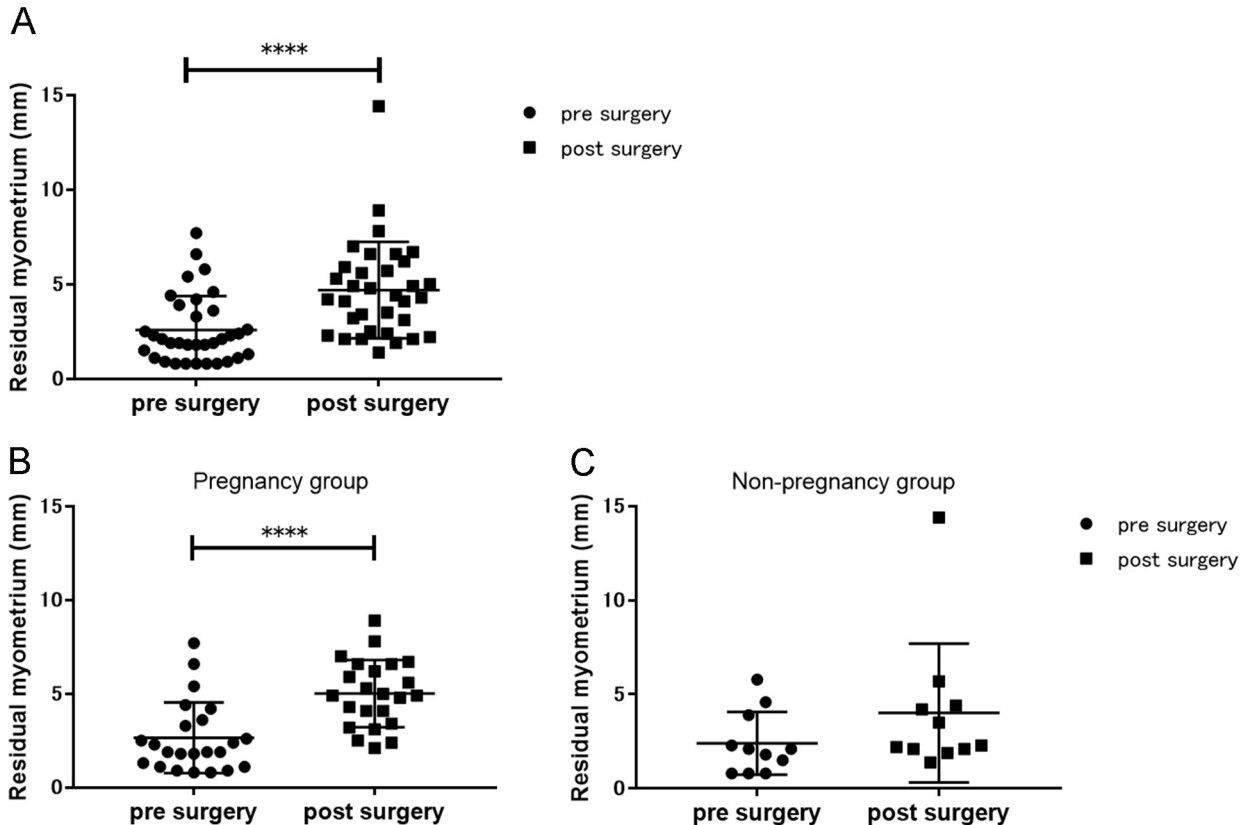

**Fig 2. Graphical representations of the residual myometrial thickness before and after hysteroscopic surgery.** The pre- and postoperative residual myometrial thicknesses of (A) the entire study population, (B) pregnant women, and (C) non-pregnant women. Significant differences between pre- and postoperative measurements were detected among the entire cohort and pregnant women. ****P < 0.0001.

retrospective observational study demonstrates the safety and efficacy of hysteroscopic surgery for treating secondary infertility caused by CSS. Our results found that the pregnancy rate was high after hysteroscopic surgery and the postoperative thickening of the residual myometrium was associated with successful reproductive outcomes.

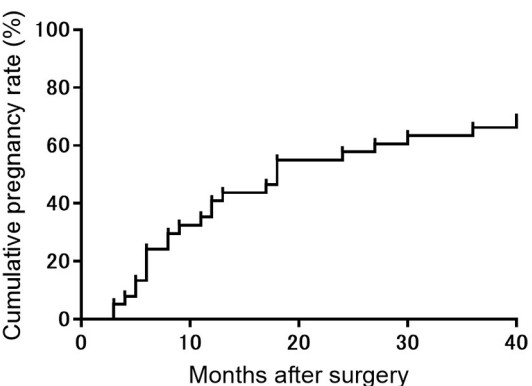

**Fig 3. Cumulative pregnancy rate after hysteroscopic surgery in women with infertility due to cesarean scar syndrome (n = 38).**

Hysteroscopic surgery in the context of CSS is mainly performed to treat abnormal uterine bleeding [11–13]; however, many recent reports have demonstrated the effectiveness of this technique for restoring fertility [8,10,14–17]. In women with CSS, infertility may arise due to abnormal uterine bleeding originating from a small hemorrhage in the CSD that interferes with implantation [8,14]. Previous reports performed hysteroscopic resection of both the superior and inferior edges of the CSD, whereas the surgical procedure used in this study resected only the inferior edge of the CSD; therefore, the present study is valuable to the field as we provide further validation of the effectiveness of hysteroscopic surgery for the treatment of infertility. Furthermore, we demonstrated the safety of our hysteroscopic surgery method, as no surgery-related complications, such as perforation, occurred, even in cases with a thin residual myometrium, and no obstetrical complications, such as uterine rupture, occurred during pregnancy. The first prospective study on the effectiveness of hysteroscopic surgery was carried out by Gubbini et al. [16] who reported that among 41 patients, no complications were noted during the perinatal period after hysteroscopic surgery. Thus, the minimally invasive technique of hysteroscopy is a safe treatment, both intraoperatively and postoperatively, in regard to subsequent pregnancy.

Regarding the main reason of infertility in participants, 68% of women became pregnant spontaneously in their prior pregnancy (S1 Table). However, they became infertile after cesarean section regardless of whether various treatments, including IVF, were conducted. On the other hand, 32% of women became pregnant by assisted reproductive technology (ART) in their prior pregnancy. Therefore, ART was performed after cesarean section during the long period; however, these patients could not become pregnant. Furthermore, abnormal uterine bleeding or liquid pooling in the CSD or uterine cavity was an obvious abnormal finding associated with infertility. Taken together, we speculate that the infertility observed in these participants was caused by CSS. However, we considered that CSS may not have been the sole cause of infertility in these patients, because around half of the patients also had endometriosis.

Although the study population had CSS-related infertility, we treated endometriosis and the area of the CSD because endometriosis is a well-known cause of infertility [18,19]. Interestingly, laparoscopic investigation revealed that around half of the patients in the present study had endometriosis in the peritoneal cavity. In the general population, the frequency of endometriosis is around 10%; however, there was a higher rate of endometriosis in present study [18]. Several reports on the presence of endometriosis in patients with CSD [17,20] are in agreement with our detection of the blueberry spot in one patient with CSD, supporting the potential association between CSS and endometriosis. In addition, the interval between surgery and pregnancy was within 2 years for most patients in the present study, and endometriosis usually recurs within 2 years of surgery [18,19]. Therefore, we suggest that this surgery may provide a sufficient "endometriosis-free" window to enable conception.

A laparoscopic retractor was sometimes useful during cauterization of the CSD with a ball electrode because it can be difficult to access the inner wall of the diverticulum due to large defects, especially when they are located on the lateral side. In the present study, the uterus was moved to the left side using laparoscopic forceps when defects were located on the right lateral side, and the side of the CSD was gently pressed from the outside of the uterus using Endo Peanut® (Medtronic, MN, USA). Therefore, hysteroscopic treatment with laparoscopy is a safe option for the treatment of infertility; specifically, laparoscopy can be beneficial during hysteroscopic surgery, especially in cases with large defects.

We have previously reported that RMT increases following our procedure of hysteroscopy [10], which is supported by the present study and a previous study [21]. Both reports used a roller ball electrode to electrocauterize the bottom of the diverticulum, with no resection of the bottom in the diverticulum. In contrast, another study that resected CSD scar tissue did not

identify changes in RMT after hysteroscopic resection [17]. Therefore, different methods of hysteroscopy may lead to varying results. Although a consensus statement from the global congress on hysteroscopy scientific committee appealed that the laparoscopic approach should be favored if the myometrial thickness is less than 3 mm, Gubbini et al. proposed that it could still be disputable to consider the indication of hysteroscopic surgery [22,23]. The mode of delivery following resectoscopic surgery in this study was planned cesarean section in all cases. Although cesarean section can result in several complications for the mother and baby, we considered the mode of delivery was better than trial of labor after cesarean section because there was no evidence of the risk of uterine rupture following hysteroscopic surgery for isthmocele [24,25].

This study has several limitations that should be considered. First, endometriosis treatment affected subsequent conception. However, the presence of endometriosis before surgery did not influence the rate of conception after surgery. We also suspect that endometriosis treatment did not affect conception following hysteroscopic surgery. Second, although there was a significant change in postoperative RMT in the pregnant group, this study could not reveal the exact mechanism of thickening. We consider that the treatment of inflammatory tissue in the CSD might promote the regeneration of fibrotic tissue or elimination of pressure as liquid pools in the defect, resulting in the thickening of the residual myometrium. Liquid pools in the CSD might affect RMT thinning via pressure. We consider that improving the environment in the uterine cavity might contribute to successful conception. In the non-pregnant group, myometrial regeneration in the CSD might have been inhibited due to insufficient treatment. It is evident that there is room for further improvement in our procedure to increase the pregnancy rate after hysteroscopic surgery. Third, this study included a small patient cohort and was a short-term study. Therefore, further investigation of a larger patient population is needed to verify the safety and efficacy of this procedure for infertile women with CSS. Fourth, due to our study's non-randomized design, the contributions of hysteroscopic surgery to subsequent pregnancy were controversial in women who became pregnant after the long period following hysteroscopic surgery.

In conclusion, hysteroscopic surgery is a safe, minimally invasive treatment for the restoration of fertility in women with CSS. Our findings suggest that the thickening of the residual myometrium following hysteroscopic surgery may influence subsequent reproductive outcomes.

## Supporting information

**S1 Fig. Intraoperative image showing the blueberry spot on the surface of the cesarean scar defect.** (B) Enlargement of the area indicated with the square in (A).
(TIF)

**S1 Table. Patients included in this study.**
(XLSX)

**S1 File. Video of the procedure details.**
(MP4)

## Acknowledgments

We would like to thank Editage (www.editage.com) for English language editing.

## Author Contributions

**Conceptualization:** Shunichiro Tsuji, Takashi Murakami.

**Data curation:** Shunichiro Tsuji, Akimasa Takahashi, Asuka Higuchi.

**Formal analysis:** Shunichiro Tsuji.

**Funding acquisition:** Takashi Murakami.

**Investigation:** Shunichiro Tsuji, Akimasa Takahashi, Asuka Higuchi, Akiyoshi Yamanaka, Tsukuru Amano, Fuminori Kimura, Ayumi Seko-Nitta.

**Methodology:** Shunichiro Tsuji, Takashi Murakami.

**Project administration:** Shunichiro Tsuji, Takashi Murakami.

**Software:** Shunichiro Tsuji.

**Supervision:** Shunichiro Tsuji, Takashi Murakami.

**Validation:** Shunichiro Tsuji, Takashi Murakami.

**Visualization:** Shunichiro Tsuji.

**Writing – original draft:** Shunichiro Tsuji.

**Writing – review & editing:** Shunichiro Tsuji, Asuka Higuchi, Akiyoshi Yamanaka, Tsukuru Amano, Fuminori Kimura, Ayumi Seko-Nitta, Takashi Murakami.

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
