## [Decision Letter · Decision Letter 0]

15 Oct 2020

PONE-D-20-27415

Pregnancy outcomes after hysteroscopic surgery in women with cesarean scar syndrome

PLOS ONE

Dear Dr. Tsuji,

Thank you for submitting your manuscript to PLOS ONE. After careful consideration, we feel that it has merit but does not fully meet PLOS ONE’s publication criteria as it currently stands. Therefore, we invite you to submit a revised version of the manuscript that addresses the points raised during the review process.

We look forward to receiving your revised manuscript.

Kind regards,

Antonio Simone Laganà, M.D., Ph.D.

Academic Editor

PLOS ONE

Journal Requirements:

2. In your Methods section, please provide additional information about the participant recruitment method and the demographic details of your participants. Please ensure you have provided sufficient details to replicate the analyses such as:   

-    a description of any inclusion/exclusion criteria that were applied to participant recruitment,

-    a table of relevant demographic details,

-    a statement as to whether your sample can be considered representative of a larger population,

-    a description of how participants were recruited, and

-       descriptions of where participants were recruited and where the research took place.

3. In the ethics statement in the manuscript and in the online submission form, please provide additional information about the patient records used in your retrospective study, including: a) whether all data were fully anonymized before you accessed them; b) the date range (month and year) during which patients' medical records were accessed; c) the date range (month and year) during which patients whose medical records were selected for this study sought treatment. If patients provided informed written consent to have data from their medical records used in research, please include this information.

4. For more information on PLOS ONE's expectations for statistical reporting, please see https://journals.plos.org/plosone/s/submission-guidelines.#loc-statistical-reporting. Please update your Methods and Results sections accordingly.

5. Please include your tables as part of your main manuscript and remove the individual files. Please note that supplementary tables (should remain/ be uploaded) as separate "supporting information" files.

Additional Editor Comments (if provided):

The topic of the manuscript is interesting. Nevertheless, the reviewers raised several concerns: considering this point, I invite authors to perform the required major revisions.

Reviewers' comments:

Reviewer's Responses to Questions

**Comments to the Author**

1. Is the manuscript technically sound, and do the data support the conclusions?

Reviewer #1: No

Reviewer #2: Partly

Reviewer #3: No

Reviewer #4: Yes

Reviewer #5: Partly

2. Has the statistical analysis been performed appropriately and rigorously? 

Reviewer #1: Yes

Reviewer #2: I Don't Know

Reviewer #3: Yes

Reviewer #4: Yes

Reviewer #5: Yes

3. Have the authors made all data underlying the findings in their manuscript fully available?

Reviewer #1: Yes

Reviewer #2: Yes

Reviewer #3: No

Reviewer #4: No

Reviewer #5: Yes

4. Is the manuscript presented in an intelligible fashion and written in standard English?

Reviewer #1: Yes

Reviewer #2: Yes

Reviewer #3: Yes

Reviewer #4: Yes

Reviewer #5: Yes

5. Review Comments to the Author

Reviewer #1: This is an interesting study to assess the pregnancy outcome after treatment of cesarean scar syndrome (CSS) by hysteroscopic surgery in infertility women. However, this paper cannot be published in the present form in the journal of PLOS ONE, since there are several questions which have to be answered efficiently from the authors.

1. This is a very small and short-term study. The author enrolled 38 infertility women because of cesarean scar defect. After hysterocopic treatment, the pregnancy outcome was analysis..

2. This is not a new study. Previous studies have been reported.

Abacjew-Chmylko A, Wydra DG, Olszewska H. Hysteroscopy in the treatment of uterine cesarean section scar diverticulum: A systematic review. Adv Med Sci. 2017; 62: 230-239. 10.1016/j.advms.2017.01.004. PMID: 28500899.

Florio P, Filippeschi M, Moncini I, Marra E, Franchini M, Gubbini G. Hysteroscopic treatment of the cesarean-induced isthmocele in restoring infertility. Curr Opin Obstet Gynecol. 2012; 24: 180-186. 10.1097/GCO.0b013e3283521202. PMID: 22395067. Bhagavath B, Lindheim SR. Optimal management of symptomatic cesarean scar defects. Fertil Steril. 2018; 110: 417-418. 10.1016/j.fertnstert.2018.06.035. PMID: 30098693.

3. How do you confirmed or diagnosed the major reason of infertility in these women were because of CSS?

4. Is there other reason of infertility in these women? Such as the author mentioned that endometriosis was detected in 19 patients (52%). The incidence of endometrisos is high in these women. Therefore, so many selection biases were noted in this study.

5. What is the incidence of endometriosis in both groups?

6. Was an attempt made to normalize data before performing non-parametric statistical analysis?

7. In this study, 17 women became pregnant in the first year, 10 in the following year. This mean that nearly 37% of patient became pregnant two years after the surgery. How do you know this is the effects of hysteroscopic surgery?

8. Did all women received IVF treatment after the hysteroscopic surgery? Especially the pregnant women?

9. Please describe clearly the reason of infertility in all women.

Reviewer #2: This is a nice paper.

I have the following questions:

1) please revise the paper in terms of grammar and language

2) references should be updated

3) authors should highlight the main limitation: the non-randomized study design and mostly the very small sample size. The conclusion should be therefore softened

4) add a video of the procedure

5) what about ultrasound follow-up in women after HST surgery?

6) add more details on diagnosis of c-scar syndrome

7) add introduction definition of c-scar sydrome, c-scar defects, isthmocele etc.....

8) how many women had diagnosis of isthmocele before surgical approach?

7) add more details on prior c-section. What type of suture? what type of closure of the uterine wall?

Reviewer #3: Low power article with clearly little experience since only 38 patients in 5 years.

The criteria for choosing between hysteroscopic, laparoscopic and vaginal routes are not specified.

On the operative technical level which hysteroscope is chosen? What diameter. Why a laparoscopic control. Is a bladder dissection performed to monitor the hysteroscopic procedure? Why no outpatient surgery? Complication not described?

In the diagnosis why MRI? Place of echosonography whose sensitivity and specificity are superior?

Clearly, nothing can be deduced from this retrospective cohort.

Reviewer #4: Authors completed in this clinical study a previous analysis which was performed to determine the residual myometrium thickness after hysteroscopic treatment of uterine scar defect. Here they demonstrate that the technique is safe and efficient to reach the goal of pregnancy in patients with secondary infertility.

They added the pregnancy outcome in 38 patients treated and the MRI measure of the myometrium demonstrated that a significant thickening of the myometrium after the procedure is related to a higher pregnancy rate.

Although a major bias, as indicated by the authors, is the high prevalence of endometriosis which was treated in most of cases and therefore influencing the conclusion that the pregnancy was achieved thanks to the hysteroscopic treatment, the study is original and could be important to select patients to enroll for the procedure. As around 50% of patients could reveal a cesarean scar defect after CS and considering the prevalence of secondary infertlity in these patients, the proposed procedure could be of a significant help for patients.

A detailed description of the population is given, but as no detail is given regarding the time frame between the CS and the definition of infertility, more information for the diagnosis of infertility would be useful to exclude any other bias of the study (were male factor, tubal factor present in the two study groups ?)

Reviewer #5: I was pleased to revise the manuscript entitled “Pregnancy outcomes after hysteroscopic surgery in women with cesarean scar syndrome” (Manuscript Number: PONE-D-20-27415).

The study was approved by the Ethics Committee at Shiga University of Medical Science (approval number; R2020-039), and written informed consent was obtained from all participants.

In my honest opinion, the topic is interesting enough to attract the readers’ attention. Methodology is accurate and conclusions are supported by the data analysis. Nevertheless, authors should clarify some points.

In general, the Manuscript may benefit from some major revisions, as suggested below:

- All the text needs a minor language revision in order to improve some typos and grammatical errors.

- Abstract. I would suggest improving the abstract reporting the mode of delivery. The main concern regarding hysteroscopic treatment of isthmocele is the mode of delivery to recommend.

- Line 68. I would suggest clarifying the concept of predictors. It is unclear if the authors refer to predictors of obstetric outcomes or predictors of reproductive outcomes. I would suggest clarifying the study aim. I would suggest better describe the literature gap that the authors desire to cover.

- Lines 76-82. I would suggest improving description of study methods. Starting from the first step and describing all passages to allow precise reproduction of the study. Too much pieces of information are missed. Which was the source of data. How patients were identified. Hysteroscopic surgery is too general. How pregnancy outcomes were identified. I would suggest referring to the STROBE/RECORD guidelines to improve the manuscript. Moreover, the study design should be clearly stated.

- Is it routine for the authors’ center to perform an RMI after 2 months from surgery? Is this a prospective or retrospective study? Please report this information in the methods and abstract.

- Lines 181-184. I would suggest clarifying in what the used technique differs from other reports.

- Lines 193-203. I would suggest better discussing the impact of endometriosis on infertility referring to its etiopathogenesis. Refer to: PMID: 32046116; PMID: 31717614. Did the authors observe differences in terms of endometriosis characteristics between women who conceived and not conceived?

- I would suggest better discussing the topic of isthmocele and pregnancy referring to the two following manuscripts. It should be better discussed the role of RMT on surgical technique and mode of delivery. In this regards, pro and cons of cesarean section should be stressed, given that all the reported cases had planned cesarean section. Refer to: PMID: 29410381; PMID: 29680233; PMID: 30877907.

6. PLOS authors have the option to publish the peer review history of their article (what does this mean?). If published, this will include your full peer review and any attached files.

Reviewer #1: **Yes: **Kok-Min Seow, MD, PhD.

Reviewer #2: No

Reviewer #3: No

Reviewer #4: No

Reviewer #5: No

---

## [Author Response · Author response to Decision Letter 0]

27 Oct 2020

Responses to Reviewer #1: 

We thank the reviewer for providing suggestions and comments, which have helped us to improve our manuscript substantially. Our replies to the additional specific questions of the reviewer are as follows:

This is an interesting study to assess the pregnancy outcome after treatment of cesarean scar syndrome (CSS) by hysteroscopic surgery in infertility women. However, this paper cannot be published in the present form in the journal of PLOS ONE, since there are several questions which have to be answered efficiently from the authors.

1. This is a very small and short-term study. The author enrolled 38 infertility women because of cesarean scar defect. After hysterocopic treatment, the pregnancy outcome was analysis.

>We agree with you. Our analysis included a small number of patients and was a short-term study. Therefore, we described these points as limitations in lines 262-265 in the discussion.

2. This is not a new study. Previous studies have been reported.

Abacjew-Chmylko A, Wydra DG, Olszewska H. Hysteroscopy in the treatment of uterine cesarean section scar diverticulum: A systematic review. Adv Med Sci. 2017; 62: 230-239. 10.1016/j.advms.2017.01.004. PMID: 28500899.

Florio P, Filippeschi M, Moncini I, Marra E, Franchini M, Gubbini G. Hysteroscopic treatment of the cesarean-induced isthmocele in restoring infertility. Curr Opin Obstet Gynecol. 2012; 24: 180-186. 10.1097/GCO.0b013e3283521202. PMID: 22395067. 

Bhagavath B, Lindheim SR. Optimal management of symptomatic cesarean scar defects. Fertil Steril. 2018; 110: 417-418. 10.1016/j.fertnstert.2018.06.035. PMID: 30098693.

>You have raised an important point; however, we believe that our study has novelty. The three study you provided demonstrated the safety and efficacy of hysteroscopic surgery in women with CSS, however, these studies did not provide evidence regarding the change in residual myometrium. Our study provides new evidence for the prediction of subsequent pregnancy after hysteroscopic surgery. 

3. How do you confirmed or diagnosed the major reason of infertility in these women were because of CSS?

>Thank you for providing this inquiry. We have clarified the diagnosis of CSS in lines 80-82 in the Materials and Methods section, and have added the following information in Supporting Information Table 1: mode of prior pregnancy, period between C/S and hysteroscopic surgery, and treatments before and after hysteroscopic surgery for infertility. As indicated in the revised SI Table 1, 68% of women in this study became pregnancy spontaneously in their prior pregnancy, however, they became infertile after cesareans section regardless of what various treatments, including IVF, they received. Additionally, 32% of women became pregnancy by ART in their prior pregnancy. Therefore, ART was performed after the cesarean section; however, these patients could not become pregnant. Furthermore, abnormal uterine bleeding or liquid pooling in the CSD or uterine cavity was an obvious abnormal finding, which was associated with infertility. Taken together, we consider that these specific participants in our study had infertility due to CSS. We added these insights in lines 202-212 in discussion.

4. Is there other reason of infertility in these women? Such as the author mentioned that endometriosis was detected in 19 patients (52%). The incidence of endometriosis is high in these women. Therefore, so many selection biases were noted in this study.

>We agree with you. We cannot deny other reasons of infertility in these women; however, there was no difference of the presence of endometriosis between the pregnancy group and the non-pregnancy group (52% vs 45%, respectively). Therefore, we believe there was no bias regarding the main conclusion, which is that the thickening of the residual myometrium may be a prediction of subsequent pregnancy after hysteroscopic surgery.

5. What is the incidence of endometriosis in both groups?

>Please see Table 1. The incidence of endometriosis was 52% in the pregnancy group, and 45% in the non-pregnancy group. 

6. Was an attempt made to normalize data before performing non-parametric statistical analysis?

>We apologize for the confusion. We performed a D’Agostino-Pearson test to analyze the normality of our data. If the data were not normally distributed, a Mann-Whitney U test was performed as a non-parametric statistical analysis.

7. In this study, 17 women became pregnant in the first year, 10 in the following year. This mean that nearly 37% of patient became pregnant two years after the surgery. How do you know this is the effects of hysteroscopic surgery?

>We agree with you. As you mentioned, the contributions of hysteroscopic surgery to subsequent pregnancy were controversial in women who became pregnancy after the long period following hysteroscopic surgery due to our non-randomized study design. Therefore, we added this point as a study limitation in lines 265-268 in the Discussion.

8. Did all women receive IVF treatment after the hysteroscopic surgery? Especially the pregnant women?

>Thank you for your inquiry. We added information regarding treatment after hysteroscopic surgery in Supporting Information Table 1.

9. Please describe clearly the reason of infertility in all women.

>Thank you for your suggestion. We revised our Supporting Information Table 1 and revised the reason for infertility in this study in lines 202-212 as follows:

Regarding the main reason of infertility in participants, 68% of women became pregnant spontaneously in their prior pregnancy (S1 Table). However, they became infertile after cesarean section regardless of whether various treatments, including IVF, were conducted. On the other hand, 32% of women became pregnant by assisted reproductive technology (ART) in their prior pregnancy. Therefore, ART was performed after cesarean section during the long period; however, these patients could not become pregnant. Furthermore, abnormal uterine bleeding or liquid pooling in the CSD or uterine cavity was an obvious abnormal finding associated with infertility. Taken together, we speculate that the infertility observed in these participants was caused by CSS. However, we considered that CSS may not have been the sole cause of infertility in these patients, because around half of the patients also had endometriosis.

Responses to Reviewer #2: 

We thank the reviewer for providing constructive suggestions and comments. Our replies to the additional specific questions of the reviewer are as follows:

This is a nice paper.

I have the following questions:

1) Please revise the paper in terms of grammar and language

>Thank you for your suggestion. We asked a native speaker to check the grammar and language again. We described this point in the Acknowledgments. Changes from a grammatical point of view were highlighted in green through revised manuscript.

2) references should be updated

>We agree with you. We have added updated references, such as reference 4.

3) authors should highlight the main limitation: the non-randomized study design and mostly the very small sample size. The conclusion should be therefore softened

>We have reflected on this comment and added these points in lines 265-268 as limitations and revised the conclusion in lines 270-272.

4) add a video of the procedure

>Thank you for the suggestion. We have added a video of the procedure as a supporting information file.

5) what about ultrasound follow-up in women after HST surgery?

>This is an interesting query. Unfortunately, we do not have ultrasound follow-up data in all participants due to the retrospective nature of the study. We apologize that we could not incorporate this intriguing suggestion.

6) add more details on diagnosis of c-scar syndrome

>We agree with your suggestion. We have added more details on our diagnosis of c-scar syndrome in lines 80-82 in the Materials and Methods section.

7) add introduction definition of c-scar sydrome, c-scar defects, isthmocele etc.....

>Thank you for your suggestion. We have added this information in lines 58-64. 

8) how many women had diagnosis of isthmocele before surgical approach?

>Thank you for inquiring about this point. All women had CSD (isthmocele) before the surgical approach. We clarified this point in lines 80-82.

7) add more details on prior c-section. What type of suture? what type of closure of the uterine wall?

>You have asked an interesting question. Unfortunately, we do not have this information due to the retrospective nature of the study.

Responses to Reviewer #3: 

We appreciate the effort taken to peer review our manuscript. Our replies to the additional specific questions of the reviewer are as follows:

Low power article with clearly little experience since only 38 patients in 5 years.

The criteria for choosing between hysteroscopic, laparoscopic and vaginal routes are not specified. On the operative technical level which hysteroscope is chosen? What diameter. Why a laparoscopic control. Is a bladder dissection performed to monitor the hysteroscopic procedure? Why no outpatient surgery? Complication not described?

>We agree with you. Our study is a small and short-term study. Therefore, we described these points as limitations in lines 262-265 in the discussion. Hysteroscopic surgery was performed in all cases, and basically in our policy, laparoscopy was performed in combination. However, two patients did not undergo laparoscopic surgery due to individual reasons. We considered that the retrospective study design was one of our study limitations. Regarding the resectoscopic system, we added details in lines 95-96. The reason for simultaneous laparoscopy with hysteroscopic surgery was to monitor accidental perforation at the site of the CSD and to treat other causes of infertility, such as endometriosis, because the aim of this operation was to restore fertility. This point was described in lines 97-100. General anesthesia was needed to perform laparoscopy, therefore, patients needed to be admitted to the hospital. We described the complications in line 137. No complication was observed in all cases. 

In the diagnosis why MRI? Place of echosonography whose sensitivity and specificity are superior?

>Thank you for the valid assessment of our diagnosis using MRI. We are currently working on another study evaluating uterine peristalsis using cine MRI (not yet submitted). Of course, this is a prospective study permitted by the ethical committee at Shiga University of Medical Science. On the other hand, the aim of the current study is to assess pregnancy outcomes, however, this study did not follow an original prospective study design. We, therefore, had to re-submit our study to our ethical committee as a retrospective study. All measurements were conducted by one senior radiologist using a high-resolution monitor who was blinded to the pregnancy outcome. We emphasized that the measurements were objective. This is why we selected MRI for diagnosis. We apologize for any confusion. 

Clearly, nothing can be deduced from this retrospective cohort.

>You have raised an important point; however, we believe our study provides new evidence for the prediction of subsequent pregnancy after hysteroscopic surgery. Our retrospective analysis identified that the thickening of the residual myometrium following hysteroscopic surgery may contribute to subsequent pregnancy. We revised the last sentence in our discussion in lines 270-272 to emphasize this point.

Responses to Reviewer #4: 

We thank the reviewer for providing such important comments. We are grateful for the time and energy that was expended to review our manuscript. Our responses to the reviewer comments are as follow:

Authors completed in this clinical study a previous analysis which was performed to determine the residual myometrium thickness after hysteroscopic treatment of uterine scar defect. Here they demonstrate that the technique is safe and efficient to reach the goal of pregnancy in patients with secondary infertility.

They added the pregnancy outcome in 38 patients treated and the MRI measure of the myometrium demonstrated that a significant thickening of the myometrium after the procedure is related to a higher pregnancy rate.

Although a major bias, as indicated by the authors, is the high prevalence of endometriosis which was treated in most of cases and therefore influencing the conclusion that the pregnancy was achieved thanks to the hysteroscopic treatment, the study is original and could be important to select patients to enroll for the procedure. As around 50% of patients could reveal a cesarean scar defect after CS and considering the prevalence of secondary infertlity in these patients, the proposed procedure could be of a significant help for patients.

A detailed description of the population is given, but as no detail is given regarding the time frame between the CS and the definition of infertility, more information for the diagnosis of infertility would be useful to exclude any other bias of the study (were male factor, tubal factor present in the two study groups ?)

>We agree with you and have incorporated these suggestions in Supporting Information Table 1. We have added the following information to this table: mode of prior pregnancy, period between C/S and hysteroscopic surgery, and treatments before and after hysteroscopic surgery for infertility. 

Responses to Reviewer #5:

We thank the referee for their careful reading our manuscript and for the helpful suggestions.

 I was pleased to revise the manuscript entitled “Pregnancy outcomes after hysteroscopic surgery in women with cesarean scar syndrome” (Manuscript Number: PONE-D-20-27415).

The study was approved by the Ethics Committee at Shiga University of Medical Science (approval number; R2020-039), and written informed consent was obtained from all participants. In my honest opinion, the topic is interesting enough to attract the readers’ attention. Methodology is accurate and conclusions are supported by the data analysis. Nevertheless, authors should clarify some points.

In general, the Manuscript may benefit from some major revisions, as suggested below:

- All the text needs a minor language revision in order to improve some typos and grammatical errors.

>Thank you for your suggestion. We asked a native speaker to check the grammar and language of our manuscript again. We described this point in the Acknowledgments. Changes from a grammatical point of view were highlighted in green through revised manuscript.

- Abstract. I would suggest improving the abstract reporting the mode of delivery. The main concern regarding hysteroscopic treatment of isthmocele is the mode of delivery to recommend.

>We agree with you and have incorporated this suggestion. We revised lines 41-42 in the abstract.

- Line 68. I would suggest clarifying the concept of predictors. It is unclear if the authors refer to predictors of obstetric outcomes or predictors of reproductive outcomes. I would suggest clarifying the study aim. I would suggest better describe the literature gap that the authors desire to cover.

>We agree with you. Predictors refers to reproductive outcomes. Additionally, regarding the safety of hysteroscopic surgery, it also refers to obstetric outcomes. We have clarified the concept of predictors throughout the paper. We revised this term in lines 71, 73, 183, 272.

- Lines 76-82. I would suggest improving description of study methods. Starting from the first step and describing all passages to allow precise reproduction of the study. Too much pieces of information are missed. Which was the source of data. How patients were identified. Hysteroscopic surgery is too general. How pregnancy outcomes were identified. I would suggest referring to the STROBE/RECORD guidelines to improve the manuscript. Moreover, the study design should be clearly stated.

>We agree with you. We have revised the study population and recruitment section. We have clarified the study design and the diagnosis of CSS in lines 78-82 in the Materials and Methods section, and have added the following information in Supporting Information Table 1: mode of prior pregnancy, period between C/S and hysteroscopic surgery, and treatments before and after hysteroscopic surgery for infertility. We considered that hysteroscopic surgery was much less invasive and that laparoscopic surgery could always be employed if hysteroscopy failed, therefore, our first choice was hysteroscopic surgery. Pregnancy outcomes were identified based on patient clinical records. For patients who did not continue to attend our hospital, we confirmed their current situation by a medical information provision form from their referral hospital or by telephone. We clarified this point in lines 120-122.

- Is it routine for the authors’ center to perform an RMI after 2 months from surgery? Is this a prospective or retrospective study? Please report this information in the methods and abstract.

>Thank you for your valid assessment. We currently are working on another study evaluating uterine peristalsis using cine MRI (not yet submitted). Of course, this is a prospective study permitted by the ethical committee at Shiga University of Medical Science. On the other hand, the aim of the current study is to assess pregnancy outcome, however, this study did not have an original prospective study design. Therefore, we re-submitted our study to the ethical committee as a retrospective study. All measurements were conducted by one senior radiologist using a high-resolution monitor who was blinded to the pregnancy outcome. We emphasized that the measurements were objective. This is why we selected MRI for diagnosis. We apologize for any confusion. 

- Lines 181-184. I would suggest clarifying in what the used technique differs from other reports.

>Thank you for the advice. We have clarified the difference between our procedure and other reports in lines 189-191.

- Lines 193-203. I would suggest better discussing the impact of endometriosis on infertility referring to its etiopathogenesis. Refer to: PMID: 32046116; PMID: 31717614. Did the authors observe differences in terms of endometriosis characteristics between women who conceived and not conceived?

>You have asked an interesting question; however, we unfortunately did not assess endometriosis characteristics. In the future, we will examine additional factors, such as ion characteristics in the pathogenesis of endometriosis. We appreciate your valuable suggestion.

- I would suggest better discussing the topic of isthmocele and pregnancy referring to the two following manuscripts. It should be better discussed the role of RMT on surgical technique and mode of delivery. In this regards, pro and cons of cesarean section should be stressed, given that all the reported cases had planned cesarean section. Refer to: PMID: 29410381; PMID: 29680233; PMID: 30877907.

>We agree with you. We referenced the three reports you provided and added a discussion about both the indication of hysteroscopic surgery in cases of thin RMT and planned cesarean section in lines 239-248.

---

## [Decision Letter · Decision Letter 1]

23 Nov 2020

Pregnancy outcomes after hysteroscopic surgery in women with cesarean scar syndrome

PONE-D-20-27415R1

Dear Dr. Tsuji,

We’re pleased to inform you that your manuscript has been judged scientifically suitable for publication and will be formally accepted for publication once it meets all outstanding technical requirements.

Kind regards,

Antonio Simone Laganà, M.D., Ph.D.

Academic Editor

PLOS ONE

Additional Editor Comments (optional):

Two of the three reviewers expressed positive comments about the revised version of the manuscript. Considering this point, after a balanced evaluation of the situation, I decided to accept the manuscript for publication.

Reviewers' comments:

Reviewer's Responses to Questions

**Comments to the Author**

1. If the authors have adequately addressed your comments raised in a previous round of review and you feel that this manuscript is now acceptable for publication, you may indicate that here to bypass the “Comments to the Author” section, enter your conflict of interest statement in the “Confidential to Editor” section, and submit your "Accept" recommendation.

Reviewer #2: All comments have been addressed

Reviewer #4: (No Response)

Reviewer #5: All comments have been addressed

2. Is the manuscript technically sound, and do the data support the conclusions?

Reviewer #2: Yes

Reviewer #4: No

Reviewer #5: Yes

3. Has the statistical analysis been performed appropriately and rigorously? 

Reviewer #2: Yes

Reviewer #4: Yes

Reviewer #5: Yes

4. Have the authors made all data underlying the findings in their manuscript fully available?

Reviewer #2: Yes

Reviewer #4: No

Reviewer #5: Yes

5. Is the manuscript presented in an intelligible fashion and written in standard English?

Reviewer #2: Yes

Reviewer #4: Yes

Reviewer #5: Yes

6. Review Comments to the Author

Reviewer #2: Authors have address all comments.

i am therefore happy with the revised version of the manuscript.

Reviewer #4: The request to provide more information of the patient characteristics was important to clarify if any variable interfering with infertility was present and to understand if the sole operative hysteroscopy was the definitive intervention by which the patient became pregnant.

Depending on the provided data this goal has not been reached, as no data have been given regarding the cause of infertility. How could he Authors say that “taken together, we speculate that the infertility observed in these participants was caused by CSS” if no detail regarding semen, tubal factor, ovarian function is provided?

Moreover, and more importantly, we are now informed that only in 2/3 of patients pregnancy was spontaneous, and 1/3 conceived by IVF. These is a bias of dramatic importance, as the prior and second pregnancy was therefore reached sometime spontaneously and in other cases by ART. We do not have information regarding the “first” or the “second” infertility. It is possible, for example, that the woman became pregnant by IVF in both cases because of male factor or other pathology of reproduction (even endometriosis which was detected in half of cases), and not because of the hysteroscopic treatment. A not-debatable study design should have included only patients with spontaneous pregnancy, who underwent a cesarean section, and a secondary infertility without any infertility factor, but the uterine scar.

Reviewer #5: I was pleased to revise the manuscript entitled “Pregnancy outcomes after hysteroscopic surgery in women with cesarean scar syndrome” (Manuscript Number: PONE-D-20-27415R1).

The study was approved by the Ethics Committee at Shiga University of Medical Science (approval number; R2020-039), and written informed consent was obtained from all participants.

In my honest opinion, the topic is interesting enough to attract the readers’ attention. Moreover, the authors addressed all the suggested revisions, and I appreciated the manuscript improvement.

7. PLOS authors have the option to publish the peer review history of their article (what does this mean?). If published, this will include your full peer review and any attached files.

Reviewer #2: No

Reviewer #4: No

Reviewer #5: No

---

## [Editor Report · Acceptance letter]

25 Nov 2020

PONE-D-20-27415R1 

Pregnancy outcomes after hysteroscopic surgery in women with cesarean scar syndrome 

Dear Dr. Tsuji:

I'm pleased to inform you that your manuscript has been deemed suitable for publication in PLOS ONE. Congratulations! Your manuscript is now with our production department. 

Kind regards, 

on behalf of

Dr. Antonio Simone Laganà 

Academic Editor

PLOS ONE